# Phosphatidylserine Supplementation as a Novel Strategy for Reducing Myocardial Infarct Size and Preventing Adverse Left Ventricular Remodeling

**DOI:** 10.3390/ijms22094401

**Published:** 2021-04-22

**Authors:** David Schumacher, Adelina Curaj, Mareike Staudt, Franziska Cordes, Andreea R. Dumitraşcu, Benjamin Rolles, Christian Beckers, Josefin Soppert, Mihaela Rusu, Sakine Simsekyilmaz, Kinan Kneizeh, Chrishan J. A. Ramachandra, Derek J. Hausenloy, Elisa A. Liehn

**Affiliations:** 1Institute for Molecular Cardiovascular Research (IMCAR), RWTH Aachen University, 52074 Aachen, Germany; dschumacher@ukaachen.de (D.S.); acuraj@ukaachen.de (A.C.); mstaudt@ukaachen.de (M.S.); franziska.neweling@uk-koeln.de (F.C.); dumitrascu.andre@yahoo.com (A.R.D.); jsoppert@ukaachen.de (J.S.); mrusu@ukaachen.de (M.R.); sakine@gmx.de (S.S.); 2Institute of Experimental Medicine and Systems Biology, RWTH Aachen University, 52074 Aachen, Germany; 3Department of Anesthesiology, University Hospital, RWTH Aachen University, 52074 Aachen, Germany; 4Department of Geriatric Medicine, University Hospital, RWTH Aachen University, 52074 Aachen, Germany; 5Department of Hematology, Oncology, Hemostaseology and Stem Cell Transplantation, Faculty of Medicine, RWTH Aachen University, 52074 Aachen, Germany; brolles@ukaachen.de; 6Institute of Immunology, Faculty of Medicine, RWTH Aachen University, 52074 Aachen, Germany; 7Department of Intensive Care and Intermediate Care, University Hospital, RWTH Aachen University, 52074 Aachen, Germany; cbeckers@ukaachen.de; 8Department of Cardiology, Angiology and Intensive Medicine, University Hospital Aachen, 52074 Aachen, Germany; kkneizeh@ukaachen.de; 9Cardiovascular & Metabolic Disorders Program, Duke-National University of Singapore Medical School, Singapore 169857, Singapore; chrishan.ramachandra@nhcs.com.sg (C.J.A.R.); derek.hausenloy@duke-nus.edu.sg (D.J.H.); 10National Heart Research Institute Singapore, National Heart Centre Singapore, Singapore 169609, Singapore; 11Yong Loo Lin School of Medicine, National University Singapore, Singapore 169857, Singapore; 12The Hatter Cardiovascular Institute, University College London, London WC1E 6HX, UK; 13Cardiovascular Research Center, College of Medical and Health Sciences, Asia University, Taichung 41354, Taiwan; 14Human Genetic Laboratory, University for Medicine and Pharmacy, 200642 Craiova, Romania

**Keywords:** L-α-Phosphatidyl-L-serine, phosphatidylserine, myocardial infarction, cardio-protection, preconditioning, inflammation

## Abstract

Phosphatidylserines are known to sustain skeletal muscle activity during intense activity or hypoxic conditions, as well as preserve neurocognitive function in older patients. Our previous studies pointed out a potential cardioprotective role of phosphatidylserine in heart ischemia. Therefore, we investigated the effects of phosphatidylserine oral supplementation in a mouse model of acute myocardial infarction (AMI). We found out that phosphatidylserine increases, significantly, the cardiomyocyte survival by 50% in an acute model of myocardial ischemia-reperfusion. Similar, phosphatidylserine reduced significantly the infarcted size by 30% and improved heart function by 25% in a chronic model of AMI. The main responsible mechanism seems to be up-regulation of protein kinase C epsilon (PKC-ε), the main player of cardio-protection during pre-conditioning. Interestingly, if the phosphatidylserine supplementation is started before induction of AMI, but not after, it selectively inhibits neutrophil’s activation, such as Interleukin 1 beta (IL-1β) expression, without affecting the healing and fibrosis. Thus, phosphatidylserine supplementation may represent a simple way to activate a pre-conditioning mechanism and may be a promising novel strategy to reduce infarct size following AMI and to prevent myocardial injury during myocardial infarction or cardiac surgery. Due to the minimal adverse effects, further investigation in large animals or in human are soon possible to establish the exact role of phosphatidylserine in cardiac diseases.

## 1. Introduction

Long term prognosis for patients with acute myocardial infarction (AMI) depends on residual heart function. Despite the significant development in treatment for ischemic heart diseases in the last decade, cardiac mortality remains the number one cause of death in developed countries. New treatments are needed to reduce myocardial infarct size and post-infarct adverse left ventricle (LV) remodeling in order to prevent heart failure and improve clinical outcomes in patients presenting with acute AMI. Infarct size and resultant cardiac function are the major determinants of post-infarct heart failure [1,2], a condition which exerts a huge global burden on healthcare and economic resources [3,4].

Although it is possible to protect the heart against AMI using endogenous cardioprotective interventions by applying brief cycles of ischemia-reperfusion to the heart, termed ‘ischemic conditioning’ [1], the translation of this cardioprotective strategy into the clinical setting has failed to improve patient outcomes in AMI patients in the CONDI-2/ERIC-PPCI trial [5]. Other strategies to improve cardiomyocytes survival under hypoxic conditions, such as targeting the mitochondrial permeability transition pore [6], fission and fusion proteins [7], ion channels [8], or endogenous ischemic conditioning signaling pathways [9] have been intensively studied in experimental settings. However, the clinical translation attempts have been hugely disappointing [10]. Therefore, new treatment strategies are needed to reduce AMI size and prevent heart failure following AMI.

In this regard, we have previously found changed phosphatidylserine fraction in the heart of Cxcr4 heterozygotes mice as a possible adaptation to chronical ischemia [11]. However, if phosphatidylserine can play a role in acute setting of AMI, as well as possible underlying mechanisms, is completely unknown. Phosphatidylserine is a crucial component of the plasma membrane of all eukaryote cells, is involved in the activation of many protein kinases and is essential for many cellular functions [12,13,14]. Phosphatidylserine supplementation has been reported to improve neurocognitive function of older patients with cognitive disorders and Alzheimer’s disease [15,16,17,18,19,20,21]. Furthermore, phosphatidylserine has proven to have ergogenic properties, improving the regeneration of skeletal muscle, reducing oxidative stress and modulating the endocrine response during excessive sport, with no major side-effects [22,23,24,25,26]. Phosphatidylserine has been shown to inhibit the production of pro-inflammatory cytokines [27,28,29] and promote anti-inflammatory pathways at the cellular level [30]. Whether phosphatidylserine supplementation is also beneficial in terms of reducing infarct size and preventing heart failure following AMI is not known. In this study, we evaluate the cardioprotective effects of phosphatidylserine supplementation in a mouse model of AMI, and investigate the mechanisms underlying these beneficial effects.

## 2. Results

### 2.1. Effects of Phosphatidylserine Supplementation on Cardioprotection

To evaluate the cardioprotective effects of phosphatidylserine treatment, a model of acute myocardial ischemia-reperfusion injury was used (Figure 1A). To omit possible long-time effects of inflammation and remodeling, the area at risk (AAR) and infarcted area were analyzed at 24 h after AMI induction. Phosphatidylserine treatment was able to increase the survival of cardiomyocytes exposed to ischemia in infarcted area (11.72 ± 3.63% from AAR in treated group vs. 23.75 ± 7.49% from AAR in control, *p* = 0.0437), as showed by Evans-blue and tetrazolium staining (Figure 1B,C). Area at risk was similar in both treated and untreated group (38.69 ± 7.42% in treated mice vs. 39.04 ± 6.00% in control group).

Since phosphatidylserines are always related to apoptosis, to exclude an effect of phosphatidylserine supplementation on apoptosis induction, we have performed TUNEL staining at 24 h after AMI to analyze apoptosis in vivo. The TUNEL staining did not show any difference between the phosphatidylserine treated and untreated mice (Figure 2A). This result was confirmed by a RT-PCR for BAX, which did not reveal any difference between treated and untreated mice (Figure 2B).

Mechanistically, we found a significant elevated myocardial expression of Protein kinase C type epsilon (PKC-ε) after phosphatidylserine treatment pre- and post-AMI (1.992 ± 0.54 relative expression, *p* = 0.0150 vs. control), as well as in the group that received phosphatidylserine only after AMI (1.838 ± 0.66 relative expression, *p* = 0.0296 vs. control), when compared to those of control group (0.96 ± 0.29 relative expression, Figure 2C). PKC-ε is a key player in cardio-protection, in the context of preconditioning, which regulates cardiomyocyte metabolism and confers mito-protection [31,32].

To further evaluate the effect of phosphatidylserine on cardio-protection, we isolated murine neonatal cardiomyocytes and treated them with phosphatidylserine during normal or hypoxic culture conditions. Interestingly, phosphatidylserine treatment of neonatal cardiomyocytes improved cell viability in a dose-dependent manner (Figure 2D). AlamarBlue^®^ uptake, as a measure of cell viability was improved with higher dosis, but not at lower dosis of phosphatidylserine in medium. Similarly, treatment with phosphatidylserine did not change mRNA expression of the pro-apoptotic protein, Bax (Figure 2E) and increased mRNA expression of the anti-apoptotic protein, Bcl2 (Figure 2F) in a dose-dependent manner (22.89 ± 7.63 relative expression after 10 µg PS treatment vs. 1.37 ± 0.78 relative expression in control group in normoxic condition, 32.51 ± 8.29 relative expression in after 10 μg PS treatment vs. 1.56 ± 0.94 relative expression in control group in hypoxic condition, *p* = 0.0012). We have found that phosphatidylserine-treatment resulted in a dose-dependent upregulation of cardioprotective mediators, such as PKC-ε (1.49 ± 0.22 relative expression after 10 µg PS treatment vs. 0.88 ± 0.22 relative expression in control group in normoxic conditions, 1.13 ± 0.12 relative expression in after 10 µg PS treatment vs. 0.69 ± 0.26 relative expression in control in hypoxic conditions, *p* = 0.0143, Figure 2G), heme oxygenase (HMOX)-1 (321.86 ± 117.41 relative expression after 10 µg PS treatment vs. 6.98 ± 10.07 relative expression in control group in normoxic conditions, 333.41 ± 82.73 relative expression in after 10 µg PS treatment vs. 18.50 ± 13.86 relative expression in control in hypoxic conditions, *p* < 0.0001, Figure 2H) and hypoxia inducible factor (HIF)-1α (22.80 ± 14.51 relative expression after 10 µg PS treatment vs. 1.65 ± 1.27 relative expression in control group in normoxic conditions, 55.56 ± 45.56 relative expression in after 10 µg PS treatment vs. 1.75 ± 0.90 relative expression in control in hypoxic conditions, *p* = 0.0097, Figure 2I), when compared to control. Interestingly, it seems that treatment of the cells with 100 µg phosphatidylserine induced a decrease in all protective genes, which can be in part of due to a mild cytotoxic effect of phosphatidylserine.

### 2.2. Effects of Phosphatidylserine Supplementation on Healing after MI and Heart Function

To analyze the long-time effects of phosphatidylserine supplementation on inflammation and remodeling, we have used a model of left anterior descending coronary artery permanent ligation, in which the effect on inflammatory and remodeling processes is more pronounced (Figure 3A). Phosphatidylserine supplementation reduced significantly infarct size and increased left ventricular ejection fraction (EF) in treated groups when compared to control (Figure 3B,C and Table 1). However, fractional shortening (FS) and left ventricular dilatation as measured by end diastolic diameter of left ventricle after AMI was reduced in the group treated with phosphatidylserine pre- and post-AMI, but not in the group that received phosphatidylserine only after AMI, when compared to control (Table 1). This can be in part due to the fact that the lack of the baseline parameters makes it difficult to highlight mild differences.

### 2.3. Effects of Phosphatidylserine Supplementation on Inflammation

As possible mechanisms of improved healing and remodeling after AMI, we have analyzed the effect of phosphatidylserine on the inflammation [11]. Unexpectedly, the number of neutrophils was significantly increased at day 4 (8.45 ± 8.46 cell/power field), and 7 (7.37 ± 8.46 cells/power field) post-AMI in the infarction area of mice treated with phosphatidylserine pre- and post-AMI, as well as in the group that received phosphatidylserine only after AMI (11.37 ± 6.74 neutrophils/power field at day 4 and 7.81 ± 3.80 neutrophils/power field at day 7), when compared to control (4.52 ± 2.04 neutrophils/power field at day 4 and 1.41 ± 0.92 cells/power field at day 7, *p* < 0.05, Figure 4A). However, we found decreased histone H3cit staining, used to assess formation of neutrophils extracellular traps (NETs), in the group treated with phosphatidylserine pre- and post-AMI, but not in the group that received phosphatidylserine only after AMI (Figure 4B). Further, mRNA expression of IL-1ß, a known neutrophil activation marker, was significantly decreased after phosphatidylserine treatment pre- and post-AMI (0.34 ± 0.28 relative expression, *p* = 0.0225 vs. control), but not in the group that received phosphatidylserine only after AMI (0.71 ± 0.46 relative expression), when compared to those of control group (1.00 ± 0.35 relative expression, Figure 4C), which can in part explain the differences in echocardiographic measurements between the groups. Thus, despite increased neutrophils recruitment, their activation seems to be impaired by phosphatidylserine treatment. To demonstrate this fact, we have isolated neutrophils from bone marrow of mice and activated them using tumor necrosis factor (TNF)-α, to mimic the post-infarction conditions. Phosphatidylserine treatment inhibited TNF-α induced activation of isolated neutrophils by reducing the expression of the activation marker CCR3 [33,34,35,36] (11.68 ± 34.50% vs. 57.32 ± 14.87% increase in CCR3 relative expression compared with control, *p* = 0.0139, Figure 4D,E).

We did not observe any differences in the infiltration of inflammatory macrophages (MPO^+^Mac3^+^, Figure 4F) or reparatory macrophages (MPO^−^Mac3^+^, Figure 4G) after phosphatidylserine treatment compared with the control in the scar, as indicated by immunofluorescence staining.

### 2.4. Effects of Phosphatidylserine Supplementation on Remodeling

Since fibrotic processes are important in scar remodeling but also in development of heart failure after AMI, we analyzed the effect of phosphatidylserine on myofibroblasts and collagen formation. Despite a trend in reduced myofibroblasts number, 7 days after AMI in treated mice, there were no significant differences between the groups (Figure 5A). Transforming growth factor (TGF)-β1, responsible for the switch to anti-inflammatory processes during healing after AMI [2], seems to be decreased by phosphatidylserine treatment pre- and post-AMI 7 days after AMI (85.54 ± 56.41 TGF-β1 positive cells/power field, *p* < 0.01 vs. control), but not in the group that received phosphatidylserine only after AMI (127.64 ± 46.65 TGF-β1 positive cells/power field), when compared to those of control group (163.08 ± 64.85 TGF-β1positive cells/power field, Figure 5B). Previously, we showed that TGF-β1 production seems to be proportional to the inflammatory processes and less to remodeling after AMI [37]. Since the inflammatory processes are reduced by phosphatidylserine supplementation pre- and post-AMI, but not in the group receiving phosphatidylserine supplementation after onset of AMI, the TGF-β1 production is also reduced only in the group treated with phosphatidylserine pre- and post-AMI and not in the group treated only after onset of AMI. Nevertheless, there is no difference in total collagen expression among all groups (Figure 5C).

## 3. Discussion

In this study we report for the first time the cardioprotective effects of phosphatidylserine supplementation in terms of reducing infarct size and improve the heart function after AMI. Although it has already been shown that intravenously administrated phosphatidylserine-loaded liposomes [38] had beneficial effects in a model of AMI, no mechanisms were investigated in this study.

Phosphatidylserine supplementation before and after induction of AMI seems to activate cardioprotective mechanisms associated with up-regulation of PKC-ε [39,40], HMOX1 [41,42], and HIF-1α [43,44,45], known to be important mediators of ischemic preconditioning. Further, phosphatidylserine supplementation reduced neutrophils activation, formation of NETs and thereby decreased the inflammatory response after AMI. Thus, the cardioprotective effects appear to be dual, with cytoprotective effects and anti-inflammatory effects.

Phosphatidylserine supplementation after onset of AMI seems to also induce cardio-protection, followed by reduction in infarction size and improved heart function. However, no reduction in anti-inflammatory processes was observed in this setting, which can explain in part why these mice still have increased diastolic volume, as a surrogate marker for ventricular dilatation. Nevertheless, this represents an important clinical setting, which should be analyzed more in detail in other models, such as large animals or even human. Our data suggests that phosphatidylserine might even be efficient if ingested orally after onset of AMI.

Phosphatidylserine supplementation has already shown benefits in patients with cognitive disorders [15,17,18,19] and in regeneration after physical activity [22,23,24,25,26] in clinical settings. Phosphatidylserine is commercialized and can be administrated without prescription. As side effect only mild gastrointestinal side-effects by high supplementation dosage were reported [22,23,24,25,26]. Due to its proven safety, phosphatidylserine supplementation represents a viable candidate as a cardio-protection strategy in AMI with immediate application in clinics. Thus, phosphatidylserine can be administrated at the onset of reperfusion, but also in cardiac surgery patients where it can be given both pre- and post-surgery.

Surprisingly, besides its direct role in AMI, phosphatidylserine reduced neutrophil activation and the subsequent pro-inflammatory reaction. Neutrophil activation is an important mechanism of cardiomyocyte death during AMI [46,47]. NETs were shown to worsen inflammation and AMI [48]. Many strategies to reduce infarct size involve reducing neutrophil infiltration, the neutrophil-induced injury and inflammation [49]. Despite the fact that the anti-inflammatory strategies have been shown to be effective in experimental animal models, their clinical translation in human studies has been challenging [50,51,52]. It is well-established that inflammatory cells are needed for a proper healing process and complete inhibition can induce significant healing defects, as well as heart rupture [37,53]. A potential solution would be to selectively inhibit IL-1β [54] as a main pro-inflammatory factor sustaining the neutrophil function. However, the results from clinical studies are still controversial [54]. In the present study, we found neutrophils to be present in the infarcted area even late after AMI. However, it seems that they are not activated and NETs formation, which assists neutrophil activity, was significantly impaired. Recently, metoprolol was shown to have a disruptive effect on neutrophils, suggesting that this might be an underestimated cardioprotective tool [55]. However, this is the first time that such a mechanism of protection has been shown in an experimental model. Despite the fact that we do not observe this effect when phosphatidylserine supplementation is done after onset of AMI, it still remains a viable alternative method to reduce infarct size and improve heart function after AMI.

It is already known that phosphatidylserines have a decisive role in many cells (e.g., by activating protein kinases from the inner membrane leaflet) [56]. Nevertheless, the role of phosphatidylserine in cell function is neglected by the scientific community and is currently often associated with apoptosis. Indeed, phosphatidylserine flips out to the external membrane leaflet during extreme cellular stress and apoptosis [57,58,59], creating a pattern on cell surface to be recognized by macrophages and antibodies (such as Annexin 5) [60,61,62]. Therefore, to exclude a correlation between the phosphatidylserine supplementation and apoptosis rate, we investigated apoptosis and genes related to apoptosis after phosphatidylserine supplementation and did not find any effect. This is probably due to the fact that not the amount of phosphatidylserine determine the apoptosis in the cell, but the location of phosphatidylserine and the pattern build on the membrane surface, to be recognized by specific receptors and antibodies [63,64]. Moreover, we may deduce that phosphatidylserine supplementation does not affect the asymmetrical distribution of the phospholipids in cell membrane to mimic apoptosis effects in an improper context.

Taken together, phosphatidylserine supplementation might be a promising therapeutic strategy to reduce infarct size and prevent the progression of heart failure. As a pre-treatment it may be beneficial in patients undergoing cardiac surgery or cardiac catheterization, where it may limit the commonly occurring peri-operative, inflammatory response, and ischemia-reperfusion injury. Therefore, phosphatidylserine supplementation might represent a promising therapeutic strategy to be tested for improving clinical outcomes following AMI.

### Study Limitations

Despite good results and limited side-effects, we consider that there are more extended investigations which should be performed before clinical application. A human pilot study for finding the proper and most effective dose would be needed to establish the value for the clinical significance. Further, we need more studies to investigate (1) how efficient is the preventive phosphatidylserine supplementation in patients presenting cardiac disease and increased risk for an AMI and (2) why is the phosphatidylserine supplementation less protective if given after the onset of AMI. Moreover, detecting the eventual side-effects depending on different pathologies, for example kidney insufficiency or diabetes, as well as the interaction with the current clinical medication is required.

## 4. Methods

All animal experiments were performed in accordance with European legislation and approved by local German authorities (LANUV—Landesamt für Natur, Umwelt und Verbraucherschutz Nordrhein-Westfalen, approval number: AZ:84-02.04.2013.A185, approval date: 16/08/2013). All mice were housed under standardized conditions in the Animal Facility of the University Hospital Aachen (Aachen, Germany).

### 4.1. Animal Models of Acute Myocardial Infarction

C57Bl/6 wild-type male 10–12 weeks old mice (Charles River, Erkrath, Germany) were subjected to AMI, as previously described [65,66]. A model of myocardial ischemia-reperfusion was used to analyze the direct cardioprotective effects. Left descendent artery permanent ligature model was used to analyze the later effects on inflammatory and remodeling processes after AMI.

In brief, mice were anesthetized using 100 mg/kg ketamine and 10 mg/kg xylazine i.p., and were intubated and ventilated with oxygen using a mouse respirator (Harvard Apparatus, March, Germany). C57Bl/6 wild-type male 10–12 weeks old mice (Charles River, Germany) were subjected to acute myocardial ischemia-reperfusion by inducing left anterior descending coronary artery occlusion for 60 min and reperfusion for 24 h. The ribs, muscle layer, and skin incision were closed, and buprenorphine (0.1 mg/kg) was administered until full recovery. The mice were randomly assigned to 2 treatment groups: (1) control group: saline vehicle control by oral gavage once daily for one week prior to AMI and until end point; (2) PS group: phosphatidylserine (5 µg/g body weight) dissolved in saline per gavage with treatment given both one week prior and until end point 24 h post-AMI. All mice were included in the analysis, unless they died during the experiment.

In separate experiments, AMI was induced after left thoracotomy by permanent ligation of the left anterior descending coronary artery with 0/7 silk. The ribs, muscle layer, and skin incision were closed, and buprenorphine (0.1 mg/kg) was administered until full recovery. Treatment was started one week prior to AMI induction (PS group) or directly following AMI (PS after group) and continued until the end point, four weeks after AMI. The mice were randomly assigned to 3 treatment groups: (1) control group: saline vehicle control by oral gavage once daily for one week prior to AMI and four weeks following AMI; (2) PS group: phosphatidylserine (5 µg/g BW) dissolved in saline applied per oral gavage with treatment given both one week prior and four weeks post-AMI; and (3) PS post group: PS only given for four weeks following AMI to simulate the clinical situation for treatment of patients diagnosed with AMI at the time of hospital presentation. At the end of the experiment the mice were euthanized using isoflurane overdose and the hearts were excised for further experiments.

### 4.2. Echocardiography

The left ventricular heart function was determined by echocardiography performed on a small-animal ultrasound imager (Vevo 770, FUJIFILM Visualsonics, Toronto, Ontario, Canada) before and four weeks after AMI. Measurements of short and long cardiac axis were taken in B-Mode (2D-realtime) and M-Mode using a 40 MHz transducer. During the procedure, mice were anesthetized with 1–2% isoflurane. The ejection fraction (EF %), fractional shortening (FS %), cardiac output (CO L/min), heart rate (HR bpm) and left ventricular diameters (mm) were recorded and analyzed with VevoLab Software. The measurements were performed by two different investigators. Since the data was similar from both investigators, only one data set is shown in the manuscript.

### 4.3. Evans-Blue/Tetrazolium Staining

Since infarction was induced by an open-chest surgery, usual markers for detection of cardiac injury, such as troponin and creatinine kinase, could not be used. For the re-perfused AMI model, area at risk (AAR) and infarction size were measured 24 h after ischemia-reperfusion injury, according to guidelines for cardio-protection analysis in experimental models [67]. The hearts were explanted, washed with PBS, and the ligature over the left anterior descending coronary artery was renewed. Next, 200 µL Evans-blue (Sigma–Aldrich, Hamburg, Germany, E2129-10G) was perfused through the aorta. After freezing for one hour at −20 °C, the hearts were cut in 6 slices, and incubated in tetrazolium solution at 37 °C for 10 min, followed by 10% formaldehyde for 10 min. The slices were fixed between microscopic slides for imaging and further measurements such as: the total ventricular area, the normally perfused area stained in blue, injured but still viable area stained in red, and the infarcted area unstained (white area). All measurements were performed using Diskus software (Hilgers, Königswinter, Germany). AAR was calculated as the difference between the total and blue-stained area and expressed as percent of total ventricular area.

### 4.4. Histology and Immunohistochemistry

Infarct size was evaluated at 4 weeks after permanent left anterior descending coronary artery ligation. Mice were anesthetized (100 mg/kg ketamine, 10 mg/kg xylazine, i.p.) and hearts were excised, fixed in formalin and embedded in paraffin. Serial sections (10–12 sections per mouse, 400 µm apart, up to the mitral valve) were stained with Gomori’s 1-step trichrome stain (Abcam, Cambridge, UK, ab150686). The infarcted area was determined for all sections using Diskus software (Hilgers, Königswinter, Germany) and expressed as a percentage of total left ventricular volume. The 3D reconstructions were performed using MATLAB Software. Blue-stained collagen content was analyzed with Cell P Software (Olympus, Hamburg, Germany) and expressed as a percentage of the infarct area [68]. Since infarction was induced by an open-chest surgery, usual markers for detection inflammation, such as CRP could not be used. Therefore, serial sections (3 sections per mouse, 400 µm apart) at different time points after AMI (1, 4, 7, 14, and 28 days) were stained to analyze inflammatory cells, such as macrophages (Mac3, BD Pharmingen, Germany), neutrophils (MPO, Neomarkers, ThermoFisher Scientific, Langerwehe, Germany), neutrophils extracellular traps (anti-H3cit antibody, Abcam, UK) and myofibroblasts (smooth muscle actin, DAKO, Germany), and TGF-ß (TGF-ß, Abcam, Cambridge, United Kingdom). TUNEL staining was performed according to manufacturer’s protocol (In Situ Cell Death Detection Kit, Sigma–Aldrich, Darmstadt, Germany). Positive-stained cells were counted in six different fields per section and expressed as number of cells per mm^2^.

### 4.5. Cell Culture and Hypoxia Experiments with Neonatal Cardiomyocytes

Neonatal cardiomyocytes were isolated from 3 days old mice, as previously described [69]. In summary, 25–30 neonatal hearts were used to isolate cardiomyocytes. Cardiomyocytes were detached from the surrounding tissue by digestion with trypsin and collagenase. After filtration, the neonatal cardiomyocytes were taken into DMEM Medium with 20% M199 (Gibco, 31150-022, Germany), 10% horse serum, 5% FCS (neonatal calf serum), 1% Hepes 1M (Gibco, 15630-056, Germany), and 1% penicillin and streptomycin. For hypoxia experiments, 1.5 million neonatal cardiomyocytes per well were seeded into 6 well plates and cultured at 37 °C with 21% oxygen and 5% CO_2_ for 10 days until 80% confluency. One hour before hypoxia, phosphatidylserine (L-α-Phosphatidyl-L-serine, Sigma–Aldrich, Darmstadt, Germany) was added to the medium in different concentrations (1 µg/mL, 10 µg/mL, and 100 µg/mL). Afterwards, the cardiomyocytes were incubated for 5 h in hypoxia (2% oxygen, 93% nitrogen, 5% CO_2_). Finally, total mRNA was extracted either as described below or AlamarBlue^®^ assay (ThermoFisher, Langerwehe Germany, DAL1025) was performed. 10% AlamarBlue^®^ was added to each well and incubated for 4 h to assess cell viability. The reduction in AlamarBlue^®^ was measured with a photometer at the wavelength 540/630 nm.

### 4.6. mRNA Isolation and RT-PCR

Total mRNA was extracted from cultured neonatal cardiomyocytes and infarcted mouse heart tissues (Qiagen Kit, Qiagen, Hilden, Germany) and transcribed (Qiagen Kit, Qiagen, Hilden, Germany) as recommended in the suppliers protocol. Quantitative real-time PCR was performed using PowerUp SYBR Green Master Mix (ThermoFischer, Langerwehe, Germany) and ViiA™ 7 Real-Time PCR System (Applied Biosystems, ThermoFischer, Langerwehe, Germany), targeting the specific genes of Bax, PKCε, Bcl2, HMOX1, HIF1α, IL-1ß, IL-6, TNF-α, and β-actin (Table 2).

### 4.7. Neutrophil Isolation and Flow Cytometry Analysis

Neutrophils were isolated from bone marrow of 10–12 weeks old male C57Bl/6 wild-type mice (*n* = 6), as previously described [37,49]. Both femur and tibia were excised, and bone marrow was flushed out with PBS. The neutrophil fraction was isolated with Histopaque 1119/1077 (Sigma–Aldrich, Darmstadt, Germany). After isolation, neutrophils were incubated for 24 h in RPMI 1640 medium with 10% FCS and 1% Penicillin and Streptomycin with or without Phosphatidylserine 10 µg/mL (L-α-Phosphatidyl-L-Serine, P0474, Sigma–Aldrich, Darmstadt, Germany). The neutrophils were activated with TNF-α (PeproTech, Hamburg, Germany) for 30 min. Cells were fixed and stained with anti-mouse F4/80-PE (BD Pharmingen, Heidelberg, Germany), anti-mouse Ly6G-APC (BD Pharmingen, Heidelberg, Germany), anti-mouse CCR3-FITC (Biolegend, San Diego, CA, USA), anti-mouse CD11b-PE-Cy7 (BD Pharmingen, Germany). Measurement of staining intensity was performed by flow cytometry (FACS Calibur, BD Biosciences, Heidelberg, Germany). CD11b^+^ and F4/8^−^ neutrophils were gated and CCR3 expression was evaluated via BD CellQuest software. CCR3 expression is shown as relative difference to the unstimulated control.

### 4.8. Statistical Analysis

Data represent mean ± SD. Statistical analysis was performed with Prism7 software (GraphPad, San Diego, CA, USA). The means of two groups were compared with unpaired Student’s-t test, using Welch’s correction by significant variance. More than two groups were analyzed using 1-way ANOVA followed by Newman–Keuls or Tukey’s multiple comparison Test or 2-way ANOVA followed by Bonferroni’s multiple comparison Test, for more than two variable parameters, as indicated. *p*-values of < 0.05 were considered significant.

## 5. Conclusions

Phosphatidylserines mediate cardio-protection by reducing the infarction size and improving heart function following acute myocardial infarction.

Phosphatidylserines have an anti-inflammatory effect by reducing the activation of neutrophils.

Phosphatidylserines supplementation is a promising potential therapeutic approach for myocardial infarction and ischemia-reperfusion injury during percutaneous coronary intervention or cardiac surgery.

### Clinical Relevance

Phosphatidylserine supplementation represents a viable low-cost strategy to increase cardiac cell survival and to reduce the inflammatory response following acute myocardial infarction. Thus, phosphatidylserine supplementation might be a promising therapeutic strategy to reduce myocardial infarction size and prevent heart failure in patients suffering a myocardial infarction and its complications. Moreover, used as a pre-treatment in patients undergoing cardiac catheterization or cardiac surgery it may limit peri-interventional and operative myocardial injury. Therefore, phosphatidylserine supplementation might represent a novel and simple therapeutic strategy for improving clinical outcomes following myocardial infarction.

## Figures and Tables

**Figure 1 ijms-22-04401-f001:**
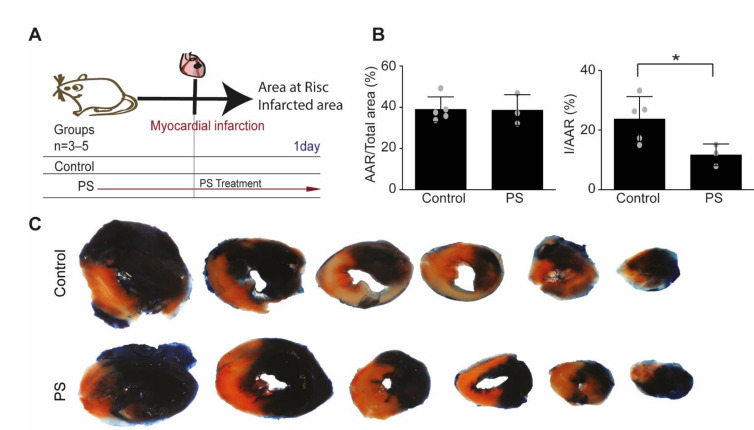
Phosphatidylserine oral supplementation before ischemia-reperfusion confers cardio-protection against acute myocardial ischemia-reperfusion injury. (**A**) Mice were randomly assigned to 2 treatment groups: (1) Control (*n* = 5): saline vehicle control by daily oral gavage one week prior to ischemia-reperfusion injury and until end of experiment; (2) PS group (*n* = 3): phosphatidylserine was given by daily oral gavage for one week prior to ischemia-reperfusion injury and until the end of experiment; Samples were collected and analyzed one day after ischemia-reperfusion to avoid inflammatory and remodeling interferences. (**B**) There was no difference in the size of the area at risk (AAR) between the two treatment groups. PS treatment reduced MI size (I) as a percentage of the AAR. (*n* = 3–5, unpaired Student’s-*t* test, * *p* < 0.05, Values ± SD). (**C**) Representative heart sections from both groups are shown (white represents infarcted myocardium, red represents viable myocardium from area of risk, blue represents non-ischemic areas).

**Figure 2 ijms-22-04401-f002:**
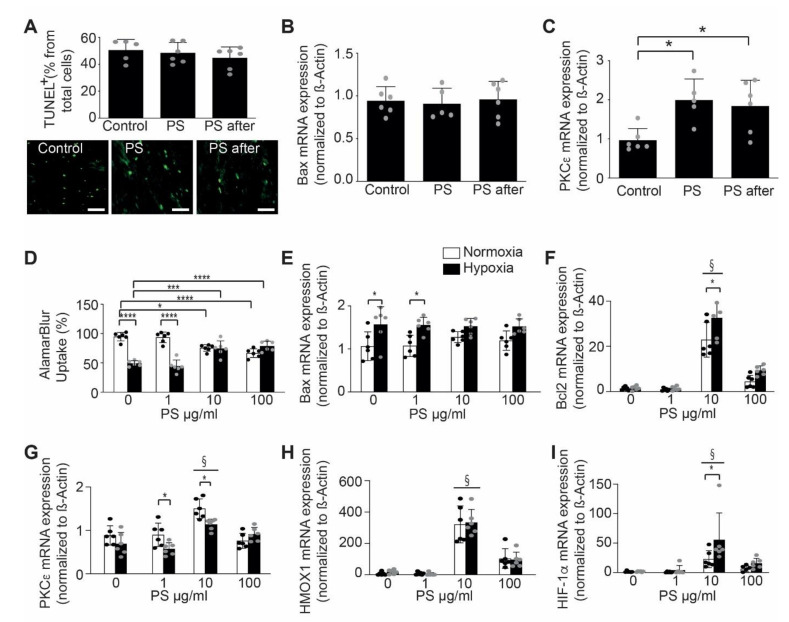
Phosphatidylserine oral supplementation effects on cardio-protection. (**A**) There was no difference in apoptotic cell death between the two groups measured by TUNEL staining of myocardial infarction in mice, untreated (Control), treated with phosphatidylserine before and after AMI (PS) or phosphatidylserine only after AMI (PS after, *n* = 5–6/group, One-way ANOVA, Tukey’s multiple comparison Test, * *p* < 0.05, Values ± SD). Lower panel shows representative images of tunnel staining. (**B**) There was no difference in Bax mRNA expression in AMI of mice, following PS treatment when comparted to control (*n* = 5–6/group, One-way ANOVA, Tukey’s multiple comparison Test, Values ± SD). (**C**) Phosphatidylserine treatment prior and post-AMI increased PKC-ε mRNA expression in AMI of mice when compared to control (*n* = 5–6/group, One-way ANOVA, Tukey’s multiple comparison Test, * *p* < 0.05, Values ± SD). (**D**) Phosphatidylserine treatment improved cell viability assessed by AlamurBlue stain following hypoxic conditions in neonatal murine cardiomyocytes when compared to control (normoxic conditions) in a dose-dependent manner (*n* = 6, Two-way ANOVA, Bonferroni’s multiple comparison Test, * *p* < 0.05, *** *p* < 0.001, **** *p* < 0.0001, Values ± SD) (normoxia-white bars, hypoxia-black bars). (**E**) Phosphatidylserine treatment protected the cardiomyocytes against apoptosis, as shown by unchanged Bax mRNA expression in treated neonatal cardiomyocytes when submitted to hypoxia (*n* = 6, Two-way ANOVA, Bonferroni’s multiple comparison Test, * *p* < 0.05, Values ± SD) (normoxia-white bars, hypoxia-black bars). (**F**) Phosphatidylserine treatment increased Bcl2 mRNA expression in neonatal cardiomyocytes treated, when compared to control (*n* = 6, Two-way ANOVA, Bonferroni’s multiple comparison Test, * *p* < 0.05, § < 0.05 vs. Control, Values ± SD) (normoxia-white bars, hypoxia-black bars). (**G**) Phosphatidylserine treatment increased PKCε mRNA expression in neonatal cardiomyocytes when compared to control (*n* = 6, Two-way ANOVA, Bonferroni’s multiple comparison Test, * *p* < 0.05, § < 0.05 vs. Control, Values ± SD) (normoxia-white bars, hypoxia-black bars). (**H**) Phosphatidylserine treatment increased HMOX1 mRNA expression in neonatal cardiomyocytes when compared to control (*n* = 6, Two-way ANOVA, Bonferroni’s multiple comparison Test, § < 0.05 vs. Control, Values ± SD) (normoxia-white bars, hypoxia-black bars). (**I**) Phosphatidylserine treatment increased HIF-1α mRNA expression of neonatal cardiomyocytes, when compared to control (*n* = 6, Two-way ANOVA, Bonferroni’s multiple comparison Test, * *p* < 0.05, § < 0.05 vs. Control, Values ± SD) (normoxia-white bars, hypoxia-black bars).

**Figure 3 ijms-22-04401-f003:**
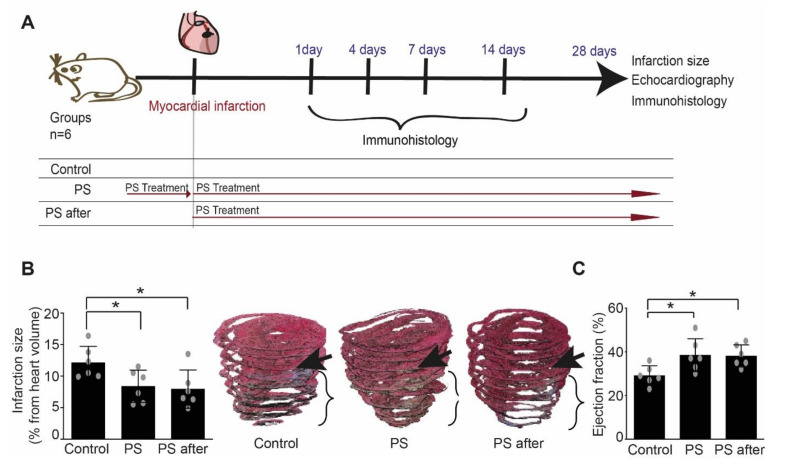
Phosphatidylserine supplementation reduced infarct size and prevented adverse post-infarct LV remodeling. (**A**) Study design: mice were randomly assigned to 3 treatment groups: (1) Control (*n* = 6): saline vehicle control by daily oral gavage one week prior to and for four weeks post-AMI; (2) phosphatidylserine group (PS, *n* = 6): phosphatidylserine was given by daily oral gavage for both one week prior and four weeks following AMI; and (3) phosphatidylserine after (PS after, *n* = 6): phosphatidylserine was given by daily oral gavage for four weeks post-AMI. Samples were collected and analyzed at the indicated time points, and infarction size and cardiac function were evaluated after 28 days. (**B**) Infarction size was significantly decreased in mice treated with phosphatidylserine when compared to control (*n* = 6, One-way ANOVA, Newman–Keuls’s multiple comparison Test, * *p* < 0.05 vs. control, Values ± SD). 3D volume reconstruction of infarcted hearts up to the mitral level from each group are shown as representative example for measuring infarction size. Arrows point out the ligature place in each heart. (**C**) Ejection fraction (%) was significantly preserved in mice treated mice with phosphatidylserine when compared to control (*n* = 6, One-way ANOVA, Newman–Keuls’s multiple comparison Test, * *p* < 0.05 vs. control, Values ± SD).

**Figure 4 ijms-22-04401-f004:**
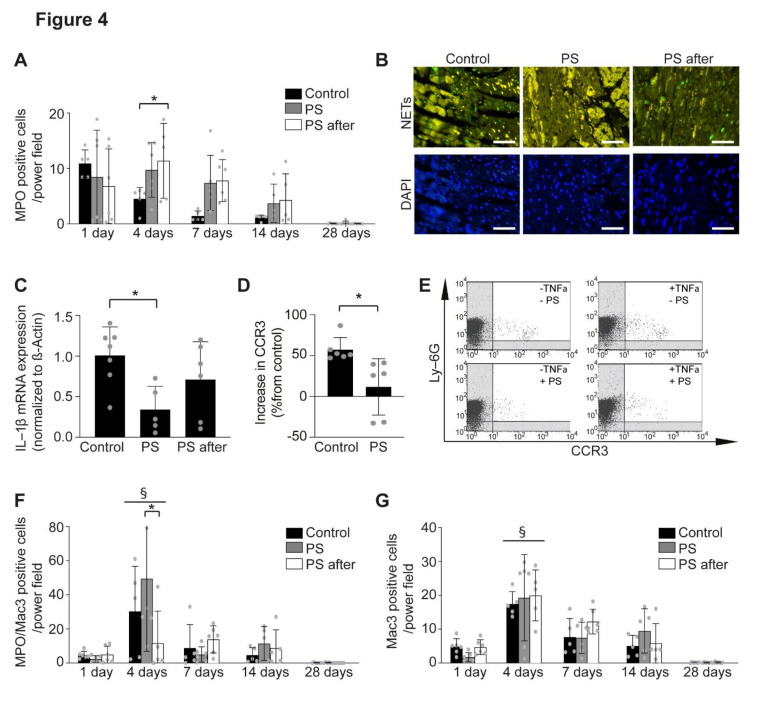
The effect of phosphatidylserine on neutrophils and inflammation after AMI. (**A**) Phosphatidylserine treatment increased the number of neutrophils in the infarcted area as assessed by MPO-positive cells at different time points after AMI, when compared to control (*n* = 5–6, Two-way ANOVA, Bonferroni’s multiple comparison Test, * *p* < 0.05, Values ± SD). (**B**) Representative images of NETs (upper panel, H3cit staining, green) and DAPI (lower panel, blue) 24 h after induction of AMI (scale bar 50 µm). (**C**) Phosphatidylserine treatment decreased IL-1ß mRNA expression in AMI, when compared to control (*n* = 5–7, One-way ANOVA, Turkey’s multiple comparison Test, * *p* < 0.05, Values ± SD). (**D**) Phosphatidylserine treatment suppressed the increase in CCR3 expression in TNF-α-stimulated or unstimulated neutrophils when compared to control (*n* = 6, unpaired Student’s-*t* test, * *p* < 0.05, Values ± SD). (**E**) Representative dot plots of FACS from neutrophils stimulated or not with phosphatidylserine. (**F**) MPO and Mac3 double positive cells in staining of infarction area by immunofluorescence at different time points after AMI in control (Control) and phosphatidylserine treated mice (PS, PS after) (*n* = 5–6, Two-way ANOVA, Bonferroni’s multiple comparison Test, * *p* < 0.05, § < 0.05 vs. 1 day, Values ± SD). (**G**) Mac3 positive cells in staining of infarction area by immunofluorescence at different time points after AMI in control (Control) and phosphatidylserine treated mice (PS, PS after) (*n* = 5–6, Two-way ANOVA, Bonferroni’s multiple comparison Test, § < 0.05 vs. 1 day, Values ± SD).

**Figure 5 ijms-22-04401-f005:**
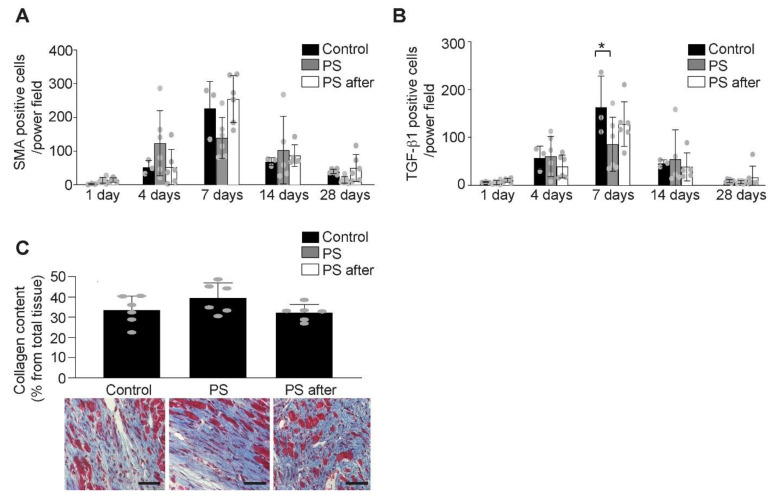
The effect of phosphatidylserine on remodeling after AMI. (**A**) The number of myofibroblasts (α-SMA positive cells) in the infarct area increased following permanent coronary ligation with no difference between the control or PS treated groups (*n* = 5–6/group, Two-way ANOVA, Bonferroni’s multiple comparison Test, Values ± SD). (**B**) TGF-β1 expression is less pronounced after phosphatidylserine treatment (*n* = 5–6/group, Two-way ANOVA, Bonferroni’s multiple comparison Test, * *p* < 0.05 vs. Control, Values ± SD). (**C**) There was no difference in myocardial collagen content (blue, Gomori staining) expressed as % from infarcted area at 28 days after AMI between control (Control) and phosphatidylserine-treated mice (PS, PS after) (*n* = 5–6/group, One-way ANOVA, Turkey’s multiple comparison Test, Values ± SD). Representative images of Gomori staining (lower panel, scale bar 50 µm).

**Table 1 ijms-22-04401-t001:** Detailed histological measurements (infarction size, heart volume) and echocardiographic parameters (EF, FS, End-Diastolic Diameter, Heart rate, Cardiac output).

Parameter	Control *n* = 6	PS *n* = 6	PS after *n* = 6	*p*
Infarct size (% from 3D reconstructed volume)	12.15 ± 2.57	8.33 ± 2.56 ^$^	7.95 ± 3.03 ^$^	* *p* = 0.0337
Heart Volume (µm^3^)	14.674 ± 2869	15.140 ± 1538	16.401 ± 2074	*p* = 0.4037
Ejection Fraction—EF (%)	29.17 ± 4.44	38.50 ± 7.53 ^$^	38.17 ± 5.07 ^$^	* *p* = 0.0225
Fractional shortening—FS (%)	19.28 ± 1.33	23.94 ± 1.13 ^$^	23.05 ± 1.07	* *p* = 0.0324
Cardiac output (ml/min)	10.34 ± 2.68	18.22 ± 3.94 ^$^	18.65 ± 4.63 ^$^	* *p* = 0.0028
End-Diastolic Diameter (mm)	4.16 ± 0.22	3.83 ± 0.19 ^$^	3.97 ± 0.21	* *p* = 0.0500
Heart rate (bpm)	384 ± 27.72	381 ± 33.83	373 ± 36.36	*p* = 0.8559

Values ± SD, ^$^ significant difference vs Control, Phosphatidylserine (PS); Ejection Fraction (EF); Fractional shortening (FS), * significant difference.

**Table 2 ijms-22-04401-t002:** Primer sequences.

Target	Forward Primer	Reverse Primer
β-actin	5’-AGCCATGTACGTAGCCATCC	5´-CTCTCAGCTGTGGTGGTGAA
PKCε	5´-GACGCCATCAAGCAACATCC	5´-TCCCGCGTAAAGTCTTGGTC
HIF1	5´-TGTTTTGAGGGCTCAGGCTC	5´-TGACATGCCACATAGCTCCC
HMOX1	5´-GAACCCAGTCTATGCCCCAC	5´-GGCGTGCAAGGGATGATTTC
Bax	5’-TGCAGAGGATGATTGCTGAC	5’-GATCAGCTCGGGCACTTTAG
BCL2	5’-AGGAGCAGGTGCCTACAAGA	5’-GCATTTTCCCACCACTGTCT
IL-1β	5’-GGATGAGGACATGAGCACCT	5’-GGAGCCTGTAGTGCAGTTGT
IL-6	5′-TCTGGAGTACCATAGCTACCTGGAGT	5′-AGCATTGGAAATTGGGGTAGGAAGGA
TNF-α	5′-GTCCCCAAAGGGATGAGAAG	5′-AGATGATCTGAGTGTGAGGG

## Data Availability

For the original date please contact the corresponding author.

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
