# Peer review of "Phosphatidylserine Supplementation as a Novel Strategy for Reducing Myocardial Infarct Size and Preventing Adverse Left Ventricular Remodeling"

_ijms, 2021, doi:10.3390/ijms22094401_

Round 1
Reviewer 1 Report
This is an interesting preclinical setting showing PS supplementation in the mice model of ischemia-reperfusion and its effects on attenuation of infarcts size and adverse LV remodeling.
Ethical statements and animal handling procedures are clearly stated.
There are some shortcomings that should be addressed, as outlined in the points below:
- Authors should include the main findings of their study in the Abstract. None of the main findings are stated in the Abstract and this should be reported.
- Results on the fibrosis and LV remodeling are less convincing than what was observed for the infarct size reduction. According to Figure 5B, it seems that the only significant difference between PS group vs. control vs. PS after MI was present at day 7 in terms of the TGF-Beta1 positive cells. Collagen tissue percentage seems the same across groups. Authors should make cautious remarks withe respect to remodeling aspect of this experimental treatment. Also, concentrations and quantification of these figures should be highlighted in the text, e.g. numerical value like 1000 vs. XXX of something, with respective p-values. Only graphical depictions are not enough in this case.
- In terms of inflammation, I think that troponin and CRP as established parameters in MI setting should have been covered in this study. This should be noted as a limitation.
- LV ejection fraction was measured by how many operators? Same/single operator? This is a subjective method, prone to inter- and -intra observer discrepancies? Could you provide evidence that you overcame this issue? If not, this should be discussed as a Limitation
- What do authors hypothesize that this drug would do in the acute setting? Because mice were pretreated with it for a relatively long time before infarct induction/ligation. Would this be feasible as a potential drug ingested orally in the early acute phase of the MI or would it be conceptually envisioned as a drug given after MI? What's the authors' position on this. More translational studies are needed in larger animal models and then in humans. This is worthy of further discussion.
Author Response
Response to Reviewer 1
This is an interesting preclinical setting showing PS supplementation in the mice model of ischemia-reperfusion and its effects on attenuation of infarcts size and adverse LV remodeling. Ethical statements and animal handling procedures are clearly stated. There are some shortcomings that should be addressed, as outlined in the points below:
Response: We thank this referee for finding our study interesting and for the further constructive comments which have helped us to improve our manuscript and make it more understandable for the reader.
Authors should include the main findings of their study in the Abstract. None of the main findings are stated in the Abstract and this should be reported.
Response: We thank you very much to point out this critical aspect. The abstract was re-written to include the main findings. Unfortunately, no number values were included due to the word limit of 200. However, we hope that the abstract is interesting enough to convince the reader to read our full manuscript.
Results on the fibrosis and LV remodeling are less convincing than what was observed for the infarct size reduction. According to Figure 5B, it seems that the only significant difference between PS group vs. control vs. PS after MI was present at day 7 in terms of the TGF-Beta1 positive cells. Collagen tissue percentage seems the same across groups. Authors should make cautious remarks with the respect to remodeling aspect of this experimental treatment.
Response: We thank you for pointing this out. We have now carefully revised the whole manuscript and made a clear delimitation between the two analyzed models in terms of remodeling (line 247-256). Thus, the differences observed between the PS and PS after group are more clearly presented in all parts of the manuscript. The remarks regarding remodeling were also carefully reviewed through all the manuscript.
Also, concentrations and quantification of these figures should be highlighted in the text, e.g. numerical value like 1000 vs. XXX of something, with respective p-values. Only graphical depictions are not enough in this case.
Response: We apologize for this inconvenient. We have now included all relevant numerical numbers in the results part for all significant experiments.
In terms of inflammation, I think that troponin and CRP as established parameters in MI setting should have been covered in this study. This should be noted as a limitation.
Response: Since in animal model the infarction is induced by opening the chest and injuring the heart from outside, troponin loose his value as measure for the infarction size. In the same way, CRP cannot be used as inflammatory marker, since it will be increased after an open chest surgery. These information are now stated in the method part as limitation of the experimental model (line 415-416 and 440-441).
LV ejection fraction was measured by how many operators? Same/single operator? This is a subjective method, prone to inter- and -intra observer discrepancies? Could you provide evidence that you overcame this issue? If not, this should be discussed as a Limitation
Response: We thank you for this observation, we have now provided this information in the method part (line 411-413). In more detail (just for reviewer): The echocardiography data were analyzed separately by two different investigators (AC and FC). Since both measurements provided similar results, we only show the data measured by one of the investigators (FC).
What do authors hypothesize that this drug would do in the acute setting? Because mice were pretreated with it for a relatively long time before infarct induction/ligation. Would this be feasible as a potential drug ingested orally in the early acute phase of the MI or would it be conceptually envisioned as a drug given after MI? What's the authors' position on this. More translational studies are needed in larger animal models and then in humans. This is worthy of further discussion.
Response: We thank you for this remark. We have now discussed in more detail the possibility of using phosphatidylserine in a clinical setting, and also extended the limitation of our study, pointing out the necessity of further investigation, mostly in large animals (line 280-287 and 343-346).
Reviewer 2 Report
Authors investigated the protective role of phosphatidylserine in AMI, focusing, in particular, on its anti-inflammatory effect. Even if the experiments are designed properly and logically connnected, some points remain not completely clear. In particular, even if Authors used two different protocols of PS treatment, one pre- and post-AMI (PS) and one only post-AMI (PS after), the results are presented and discussed as generic "PS treatment", without discriminating the two protocols. This could be confounding since the two protocols gave rise to differential results in some experiments. I suggest to Authors to distinguish the two treatments when presenting the results and review the discussion in light of the differences between the two.
In figure 1B and 3B the infarct size is quantified with two different modalities. It will be better to present the data analyzed in the same way, as % from total heart volume.
At row 241, Authors talk about "decreased mRNA expression of Bax" but in figure 2E there is no difference in Bax expression with PS treatment. The loss of statistical significance at 10 and 100 μg/ml PS between normoxia and hypoxia seems more related to a mild increase in normoxic condition.
In figures 2F-I, how Authors explain the increased expression of the relative genes both in normoxia and in hypoxia at 10 μg/ml PS? Since both 10 and 100 μg/ml PS induced a rescue in cell viability, how Authors explain that there is not a significant regulation of the expression of these genes at 100 μg/ml PS? In the figure legend, at row 261, change E in F.
At row 276, Authors talk about "preserved LVEF". Since we do not have the echocardiographic parameters at time 0 (before AMI) is better to report it as "increased compare to control". I suggest to Authors to report in the text also the differences in the cardiac output and FS. For FS and EDD, there is a significant difference only in the PS protocol and not in the PS after one. How Authors explain this? Maybe the lack of parameters at time 0 made it difficult to highlight mild differences.
In figure 4A, is the difference at 4 days significant only for the PS after? If yes, it is better to discriminate between the two protocols. Unlike what is written in the text, no significance is indicated for data at 7 days. In figure 4B, the difference in NETs between control and PS is not convincing. Are NETs decreased only in PS after protocol? If yes, indicate it in the text. Also for figure 4C, indicate that IL-1β expression is modulated only in the PS protocol. How Authors explain that NETs formation and IL-1β modulation are different between the two PS protocols?
In figure 5B, is not clear the Authors' hypothesis about the reduction in TGF-β1 positive cells at 7 days in PS protocol. Explain it more clearly referring to the specific PS protocol.
Author Response
Response to Reviewer 2
Authors investigated the protective role of phosphatidylserine in AMI, focusing, in particular, on its anti-inflammatory effect. Even if the experiments are designed properly and logically connected, some points remain not completely clear.
Response: We thank this referee to find our results properly designed and logically connected and for the constructive comments, pointed out some weaknesses of our manuscript. We have now corrected and improved all the required issues. We hope that our manuscript is now improved significantly.
In particular, even if Authors used two different protocols of PS treatment, one pre- and post-AMI (PS) and one only post-AMI (PS after), the results are presented and discussed as generic "PS treatment", without discriminating the two protocols. This could be confounding since the two protocols gave rise to differential results in some experiments. I suggest to Authors to distinguish the two treatments when presenting the results and review the discussion in light of the differences between the two.
Response: Thank you for this remark. We have now reviewed the manuscript and carefully distinguished between the two models to avoid any confusion.
In figure 1B and 3B the infarct size is quantified with two different modalities. It will be better to present the data analyzed in the same way, as % from total heart volume.
Response: The models, tissue preparing and data output used in Figures 1B and 3B are different. Figure 1B presents an early time point after ischemia/reperfusion model, performed and measured according to the guideline published in 2018 regarding experimental models of experimental ischemia/reperfusion (https://doi.org/10.1007/s00395-018-0696-8, reference 67). This aspect is now mentioned in the method part (line 418-419). Figure 3B presents a model of permanent LAD ligature for which we are only able to measure the total heart area and infarcted area at a later time point after AMI. Since the time points and models are different, we had to use different modalities to quantify.
At row 241, Authors talk about "decreased mRNA expression of Bax" but in figure 2E there is no difference in Bax expression with PS treatment. The loss of statistical significance at 10 and 100 μg/ml PS between normoxia and hypoxia seems more related to a mild increase in normoxic condition.
Response: Thank you for this observation. We have presented now the results accordingly (line 124).
In figures 2F-I, how Authors explain the increased expression of the relative genes both in normoxia and in hypoxia at 10 μg/ml PS? Since both 10 and 100 μg/ml PS induced a rescue in cell viability, how Authors explain that there is not a significant regulation of the expression of these genes at 100 μg/ml PS? In the figure legend, at row 261, change E in F.
Response: We agree with the referee that these results need an explanation, and we thank you for pointing this out. Unfortunately, the signaling pathways induced by phospholipids are currently difficult to investigate. Therefore, we can only speculate that treatment of the cells with 100 mg phosphatidylserine induced a decrease in all protective genes, which can be in part due to increased cytotoxic effect of phosphatidylserine. This is now stated in the results part (line 130-142).
At row 276, Authors talk about "preserved LVEF". Since we do not have the echocardiographic parameters at time 0 (before AMI) is better to report it as "increased compare to control". I suggest to Authors to report in the text also the differences in the cardiac output and FS. For FS and EDD, there is a significant difference only in the PS protocol and not in the PS after one. How Authors explain this? Maybe the lack of parameters at time 0 made it difficult to highlight mild differences.
Response: We agree with the referee that the lack of measurement at base line makes us now difficult to interpret the results. Therefore, we have stated this aspect as a limitation of our study (line 178-179). The PS after protocol does not have the same effect on inflammation as the pre and post-AMI protocol. This might explain the differences for FS and EDD are probably only seen in the PS after protocol. (line 280-284)
In figure 4A, is the difference at 4 days significant only for the PS after? If yes, it is better to discriminate between the two protocols. Unlike what is written in the text, no significance is indicated for data at 7 days. In figure 4B, the difference in NETs between control and PS is not convincing. Are NETs decreased only in PS after protocol? If yes, indicate it in the text. Also for figure 4C, indicate that IL-1β expression is modulated only in the PS protocol. How Authors explain that NETs formation and IL-1β modulation are different between the two PS protocols?
Response: We agree with the referee and we thank you to help us to point out our results more clear and understandable way. Indeed, we have observed the differences in neutrophil activation only in the group treated with phosphatidylserine before and after MI. We have now described this in more detail in the results part for NETs and IL-1beta (line 204-212). Unfortunately, we do not have any explanation for this, therefore we can only speculate that this aspect can explain in part the differences observed in functional parameters measured in echocardiography (line 212-213).
In figure 5B, is not clear the Authors' hypothesis about the reduction in TGF-β1 positive cells at 7 days in PS protocol. Explain it more clearly referring to the specific PS protocol
Response: Thank you to point this out. We have now explained in detail the possible mechanism of these observation (line 247-256). Since TGF-b1 seems to be proportional to the inflammatory processes and less to the remodeling [Reference 37 in our manuscript], it´s expression is probably also reduced in the group treated with PS pre- and post-AMI (considering that the inflammation is only reduced in this treatment group.
Round 2
Reviewer 1 Report
The authors have answered all my comments in a satisfactory fashion.